**Data Availability Statement:** According to Twitter agreement, we release IDs (Tweet and User ID) of the 7k mask tweets, but their content. The IDs are

# The social amplification and attenuation of COVID-19 risk perception shaping mask wearing behavior: A longitudinal twitter analysis

**Suellen Hopfer**[ORCID][1]*, **Emilia J. Fields**[1], **Yuwen Lu**[2], **Ganesh Ramakrishnan**[ORCID][3], **Ted Grover**[4], **Quishi Bai**[2], **Yicong Huang**[2], **Chen Li**[2], **Gloria Mark**[4]

1 Department of Health, Society & Behavior, Program in Public Health, University of California, Irvine, Irvine, California, United States of America, 2 Department of Computer Science, University of California, Irvine, CA, United States of America, 3 Department of Psychological Sciences, University of California, Irvine, Irvine, California, United States of America, 4 Department of Informatics, University of California, Irvine, Irvine, California, United States of America

* shopfer@hs.uci.edu

## Abstract

### Introduction

Twitter represents a mainstream news source for the American public, offering a valuable vehicle for learning how citizens make sense of pandemic health threats like Covid-19. Masking as a risk mitigation measure became controversial in the US. The social amplification risk framework offers insight into how a risk event interacts with psychological, social, institutional, and cultural communication processes to shape Covid-19 risk perception.

### Methods

Qualitative content analysis was conducted on 7,024 mask tweets reflecting 6,286 users between January 24 and July 7, 2020, to identify how citizens expressed Covid-19 risk perception over time. Descriptive statistics were computed for (a) proportion of tweets using hyperlinks, (b) mentions, (c) hashtags, (d) questions, and (e) location.

### Results

Six themes emerged regarding how mask tweets amplified and attenuated Covid-19 risk: (a) severity perceptions (18.0%) steadily increased across 5 months; (b) mask effectiveness debates (10.7%) persisted; (c) who is at risk (26.4%) peaked in April and May 2020; (d) mask guidelines (15.6%) peaked April 3, 2020, with federal guidelines; (e) political legitimizing of Covid-19 risk (18.3%) steadily increased; and (f) mask behavior of others (31.6%) composed the largest discussion category and increased over time. Of tweets, 45% contained a hyperlink, 40% contained mentions, 33% contained hashtags, and 16.5% were expressed as a question.

published in the file https://github.com/ISG-ICS/
Geolocated_USA_tweets_w_mask_IDs/blob/main/
geolocated_USA_tweets_w_mask_full%20-%
20geolocated_USA_tweets_w_mask_IDs_only.csv

**Funding:** Mark, G., Li, C., Hopfer, S. were awarded
an National Science Foundation Covid19 award
#2027254. The funders had no role in study
design, data collection and analysis, decision to
publish, or preparation of the manuscript.

**Competing interests:** No authors have competing
interests.

## Conclusions

Users ascribed many meanings to mask wearing in the social media information environment revealing that COVID-19 risk was expressed in a more expanded range than objective risk. The simultaneous amplification and attenuation of COVID-19 risk perception on social media complicates public health messaging about mask wearing.

## Introduction

On January 30, 2020, the International Health Regulations Committee of the World Health Organization declared the now recognized Coronavirus 2019 (COVID-19) a public health emergency of international concern [1]. The outbreak of severe acute respiratory syndrome Coronavirus 2 (SARS-CoV-2), the virus that causes COVID-19 respiratory disease, has spread across the globe including the United States, resulting in a staggering loss of life—more than 4 million deaths worldwide as of August 8 2021, with more than half a million deaths (616,700) in the United States [2]. In 2021, well over a year into the pandemic, the United States continues to experience COVID-19 cases and deaths, constituting a devastating public health and economic burden.

### Facemask wearing as a key COVID-19 risk mitigation measure

Facemask wearing became one of the dominant nonpharmaceutical risk mitigation strategies to prevent the spread of COVID-19 in the absence of vaccines [3] and continues to play a key role in the prevention of the highly infectious SARS-CoV-2 virus and its variants, even with the emergency authorization of three COVID-19 vaccines. Despite evidence since March 2020 that wearing facemasks can effectively reduce droplet and airborne transmission of COVID-19 risk [4, 5], the acceptability of wearing face masks in the U.S. public has varied dramatically, with low mask-wearing compliance [6, 7]. Many factors contributed to the phenomenon of noncompliance with mask wearing, including that it was not a prior norm in the United States, that COVID-19 was an unfamiliar risk, and that the Centers for Disease Control and Prevention (CDC) explicitly discouraged the public to wear masks initially. Furthermore, public health messaging initially focused on surface rather than droplet and airborne transmission, contributing to the absence of a rationale for mask wearing as a protective behavior [8]. Moreover, wearing masks became highly politicized in the United States rather than a common sense measure, in part as the result of former President Donald Trump not role-modeling mask wearing and downplaying the seriousness of the public health risk of the coronavirus [9, 10]. The psychological literature has shown that risk perception, and not objective risk assessment, is a key important driver of behavior [11–13]. Therefore, we turn to risk perception theory to explain the differential public response to mask wearing.

### COVID-19 risk and the social amplification risk framework

The social amplification risk framework (SARF) describes the dynamic social processes underlying risk perception [14, 15]. Empirical investigations have examined the public response to risk in terms of how risk perception intensifies or attenuates as a result of various amplification stations (e.g., media, social institutions, interpersonal interactions, social media) [15, 16]. The tenets of SARF view risk perception as a process in which risk signals (defined as messages about a hazard that affect people's perceptions about the seriousness or manageability of a

risk) interact with psychological, social, institutional, and cultural processes in ways that intensify or attenuate perceptions of risk and its manageability, and shape risk behavior [17]. The experience of risk is, therefore, not only an experience of a physical threat, but also the result of processes by which groups and individuals learn to ascribe new meanings to risk and create subjective interpretations of risk [15]. We define COVID-19 risk perception based on Kasperson [14, 15] as a communication process situating risk perception cognitively, but also socially, culturally, and historically. We used this definition to describe what risk signals were discussed in mask-related tweets on Twitter, which can give insight into what shapes the public response to SARS-CoV-2 risk and mask-wearing risk mitigation efforts [15, 16].

We aimed to use SARF to address the gap in research on risk that previously was dominated by a focus on technical objective constructions of risk, resulting in discrepancies between expert and lay perceptions of risk [18]. SARF focuses on how information processes, institutional structures, social group behavior, and individual responses shape the social experience of risk and on message receiver reactions to risk information contributing to risk consequences [18]. SARF acknowledges that many risks are only indirectly experienced, resulting in exposure to various risk signals having the potential to greatly influence and shape developing risk perceptions.

The conceptual risk framework SARF posits that intensification or attenuation of risk occurs in two stages: in the initial transfer of information about the risk and in the response mechanisms of society (behavioral, economic, and symbolic impacts) [15]. According to SARF, at least four information mechanisms can contribute to the intensification of risk perceptions in the initial transfer of risk information: (a) volume of information about a risk; (b) ambiguity of information or the degree to which it is disputed; (c) the extent to which information is dramatized; and (d) the symbolic connotations of the information [15, 16]. In SARF, the metaphor of amplification from classical communication theory refers to how various social agents generate, receive, interpret, and pass on risk signals. This study applied SARF to the social media context to uncover and describe how the initial transfer of risk information (as shared on Twitter) gives insight into how individuals create and ascribe meaning to Covid-19 risk and mask wearing.

Only a handful of studies have applied SARF to understand how the social media context contributes to created interpretations of risk [19–23]. Social media platforms have become an increasingly mainstream source for news and public health risk information, warranting investigation into how this context contributes to interpretations of risk [24]. Studies have examined how social media brought attention to a risk issue not covered by traditional media [19, 20], ranging from examining differential linguistic expressions of blame during the 2015 Zika crisis across languages (English, Spanish, and Portuguese) and platforms (Facebook and Twitter) [21] to organizational use of hyperlinks, images, and videos that amplify cancer risk perceptions [22]. Social media contexts are ripe for studying how individuals and stakeholders generate, receive, interpret, and pass on risk signals to understand how the response to and transformation of risk signals result in consequent adoption or rejection of risk mitigation behaviors.

## Twitter as social media risk message amplifier

Social media contexts, now a mainstream news and public health information source for U.S. adults [24], are an especially important communication context, given their widespread adoption across the United States and the world for public health risk information [25, 26]. The US leads in Twitter users, with 68.7 million users turning to the social media platform daily for their news, including COVID-19 information [26]. These interpersonal platforms with a mass

media reach play an increasingly important role in alerting people to disasters or pandemics [22, 24, 25]. The social media context consequently offers a space wherein not only are risk signals shared, but often risk messages are edited or passed on with new additional affordances and meanings ascribed to them that in turn, shape interpretation of risk and subsequent adoption of risk prevention behaviors [17]. Against this backdrop, we investigated how Twitter mask tweets described COVID-19 risk (explicitly and implicitly), how they intensified or attenuated COVID-19 risk perceptions, and how risk perceptions changed during the first 5 months of the pandemic. Our study aimed to gain a greater understanding of the factors shaping public COVID-19 risk perceptions, but also how findings can inform public health messaging strategies to increase message acceptance around mask wearing. On a theoretical level, the study aimed to advance insights into the effects of the social amplification and attenuation of risk as a result of how risk is conveyed on social media.

## Study aims

Guided by the tenets of SARF, we posed the following research questions:

RQ1: What are the different ways in which mask tweets express COVID-19 risk perceptions?

RQ2: How do expressions heighten and attenuate the salience of COVID-19 risk?

RQ3: How do these expressions change over time?

## Methods

### Data collection

The data for the current study included tweets from the microblogging platform Twitter across the first 5 months of the pandemic (January 24 through July 7, 2020). Tweets containing keywords related to the Coronavirus (see S1 List for list of keywords) were filtered and stored via the Twitter search API version 1.1 [27], which searches against the past 7 days of tweet history, and then Twitter Filtered Stream API version 1.1 [28], which filters on a real-time tweet stream. With the two APIs combined, a continuous stream of ~1% of all tweets with the given keywords were collected. COVID-19 and coronavirus tweets were collected at large from the beginning of the pandemic (January 2020). From this set of coronavirus-related tweets, we then filtered for tweets that contained geolocation information associated with the tweets and filtered for locations in the US. The sample of tweets for this study was restricted to the US and English tweets. The research team made the decision to narrow the scope of analysis on mask tweets in the US given that mask wearing was controversial particularly in the US in the early phases of the pandemic with variation in risk response across states.

In addition, from all Coronavirus-related tweets, we derived a set of 150 of the most popular news agencies and COVID-19 news-sharing accounts and filtered tweets from these accounts from the set of US geolocated tweets (see S2 List for a list of news agencies). We did so to ensure our dataset primarily reflected discussions and information sharing from citizens. We further filtered our dataset to remove tweets that only contained hashtags or user mentions (i.e., "@" or "#" prefixes) with no other words and non-English tweets. The dataset we used for this analysis was user generated content only. The dataset of tweets therefore, did not include instances where a user retweeted another tweet. This step resulted in a cleaned dataset of 592,317 tweets. These US geolocated tweets were then filtered again to narrow the focus to tweets mentioning mask-related keywords: mask, masks, facemask, or facemasks. This step resulted in a final set of 7,024 geolocated and mask-related US tweets that we used for the analysis.

## Data analysis

A manual qualitative content analysis of the 7,024 tweets reflecting 6,286 unique users was conducted using a phronetic iterative approach [29, 30]. This data analysis approach alternates between inductively analyzing data (i.e., tweet content) to identify how tweets expressed COVID-19 risk perception (labeling ideas with descriptive codes) and then using theory, i.e., SARF, to guide data analysis for how data intensified or attenuated Covid-19 risk perceptions and consequently, attitudes toward mask wearing [16]. Of the 7024 tweets, 1072 (15%) were coded as not relevant to expressing risk perception. These mask tweets were comprised of mask advertisements, tweets discussing other countries, or difficult to make sense of the tweets. Explicit and implicit expressions of risk perception were included in the codebook. Additionally, for each risk perception aspect, i.e., theme, tweets were analyzed for how they intensified or attenuated COVID-19 risk perception.

**Codebook development.**   A team of three coders met weekly across 9 months to review tweets, discuss interpretations, and label codes to develop a codebook with examples and exclusion and inclusion criteria. Code names and the codebook criteria were developed inductively and iteratively over time (see S1 Table for codebook). To ensure that emergent codes related to risk perceptions were discovered (i.e., saturation of codes), coders reviewed randomly sampled batches of 150 mask tweets from each month across the 5 months of data. Each coder reviewed and coded their sample of tweets independently, followed by the coding team meeting to discuss and compare how they labeled descriptive codes and interpreted each tweet for expression of COVID-19 risk. The coding team then iteratively developed the codebook describing how mask tweets covered topics as they related to risk. The coding team reviewed and discussed approximately 950 mask tweets to agree on interpretations of mask tweets, discussed how tweets reflected expressions of risk explicitly and implicitly, and reviewed the multiple ideas reflected in each tweet (a tweet could be labeled with up to three codes). In the codebook, example tweets were included for each identified code and articulated inclusion and exclusion criteria to discriminate and clarify interpretation of mask tweets.

**Intercoder reliability.**   Once saturation of code development was reached after covering the full timeline of the data (5 months), intercoder reliability regarding the identified codes was computed for randomly sampled and assigned batches of tweets. Fleiss' Kappa, which calculates the degree of agreement in classifying nominal codes over that which would be expected by chance, was computed for each code to assess intercoder reliability among the three coders [31, 32]. Kappa ranges from 0 (no agreement) to 1 (perfect agreement), with .61–.80 indicating substantial agreement and .81–.99 indicating almost perfect agreement [33, 34]. Initial kappa estimates were low for some codes. The coding team discussed discrepancies in tweet interpretation and clarified the codebook. New batches of 150 tweets were then coded independently to retest intercoder reliability, repeating this process until intercoder reliability of at least .60 was obtained to indicate substantial agreement among coders [32–34].

**Mask tweet frequencies and proportions.**   Descriptive statistics were computed for each identified mask risk code [6] during the 5 months of data to identify and examine mask discussion volume trends over time. External events were plotted in parallel to map associations between external events and peaks of mask tweet discussions (e.g., mask guidelines changes, stay-at-home orders, etc.).

Three phases marked changes in response to COVID-19 risk perception across the first 5 months of the pandemic (January 24 through July 7, 2020). Phase 1 included mask tweets beginning January 24, 2020, (first data collection point) through April 3, 2020, when the CDC recommended mask wearing for the asymptomatic general public. Mask tweets surged during and after April 3 in response to the guidelines, which contradicted prior explicit guidelines for

the asymptomatic public to not wear masks. Phase 2 was marked by mask tweets after April 3, 2020, including March 2020, during which the first state lockdown and school closures occurred (e.g., California), and including May, marked by the death of George Floyd on May 25 and the subsequent Memorial Day, when Black Lives Matter (BLM) protests erupted. Phase 3 included mask tweets after Memorial Day until July 7 (after the July 4 weekend), during which there were BLM protests, a staggered reopening of the economy, summer, and the steady increase in COVID-19 cases across the country.

**Data analysis of additional twitter affordances.** The dataset of tweets was filtered for presence of hyperlinks or URLs, mask hashtags, mentions, and tweets expressed as questions (interrogatory; tweets that contained a question mark). Data were additionally analyzed for how the mask hashtags qualitatively changed over time, and what figures were "mentioned" in mask tweets (e.g., political, government, and public health officials, journalists).

## Results

Data included a sample of 7,024 geo-located mask tweets from the first 5 months of the pandemic in 2020 (see Fig 1 for proportion of mask tweets across phases). Tweets were expressed as questions, observation of others, personal experiences, comments on policies, hypothetical scenarios, expressions of attitudes, and sharing of hyperlinks, photos, videos, hashtags, and mentions. Mask tweets increased over time (Fig 1) witnessing peaks and lulls with an overall trend of increasing across the first five months of the pandemic. Analysis of mask tweet content discovered six derived themes (Fig 2) in which COVID-19 risk perception was expressed by the public as: (a) severity discussions about the novel SARS-CoV-2 virus (18.0% of mask tweets, $\kappa = .64$); (b) who is at risk (26.4%, $\kappa = .64$); (c) mask effectiveness (10.7%, $\kappa = .75$); (d) mask guidelines and policies (15.6%, $\kappa = .82$); (e) political legitimizing of Covid-19 risk (18.3%, $\kappa = .74$); and (f) mask behavior of others (31.6%, $\kappa = .74$).

### Change in mask tweet expressions over time

Tweet frequency of the derived themes changed across the first 5 months of the pandemic in phases, indicating that some themes were more prevalent early on and others became more prevalent as the pandemic evolved (see Fig 3). Within the first phase of the pandemic, marked by January 24 until April 3[rd] of 2020, among mask discussions, "who is at risk" was the most discussed theme followed by severity perception discussions. Tweets about who is at risk persisted and increased in the second phase of the pandemic (through end of May) to then persist across the third phase (June and July summer) but relative to "mask behavior of others" became less of a topic. Severity discussion, which peaked initially, witnessed a modest drop as discussion topic (relative to mask behavior of others, who is at risk, politicization, and guidelines) but then steadily increased in volume after March and peaked in the third phase of the pandemic after Memorial Day and beyond through summer. The volume of mask effectiveness tweets increased moderately throughout the 5 months. Mask guidelines as a topic peaked in Phase 2 (April 3, 2020) when the CDC changed its mask guidelines and dropped thereafter. See Fig 4 for external real-world events that occurred as Twitter mask discussions and risk perception themes evolved over time. Politicization of mask wearing as a topic steadily increased across the first 5 months of the pandemic. Discussions about "mask wearing behavior of others" steadily increased over time, dramatically increasing with the onset of summer in June. Information seeking expressed in tweets as questions persisted across the first 5 months, peaking in Phase 2 during state and school lockdowns (3.6% of tweets in Phase 1, 7.3% in Phase 2, and 5.6% in Phase 3).

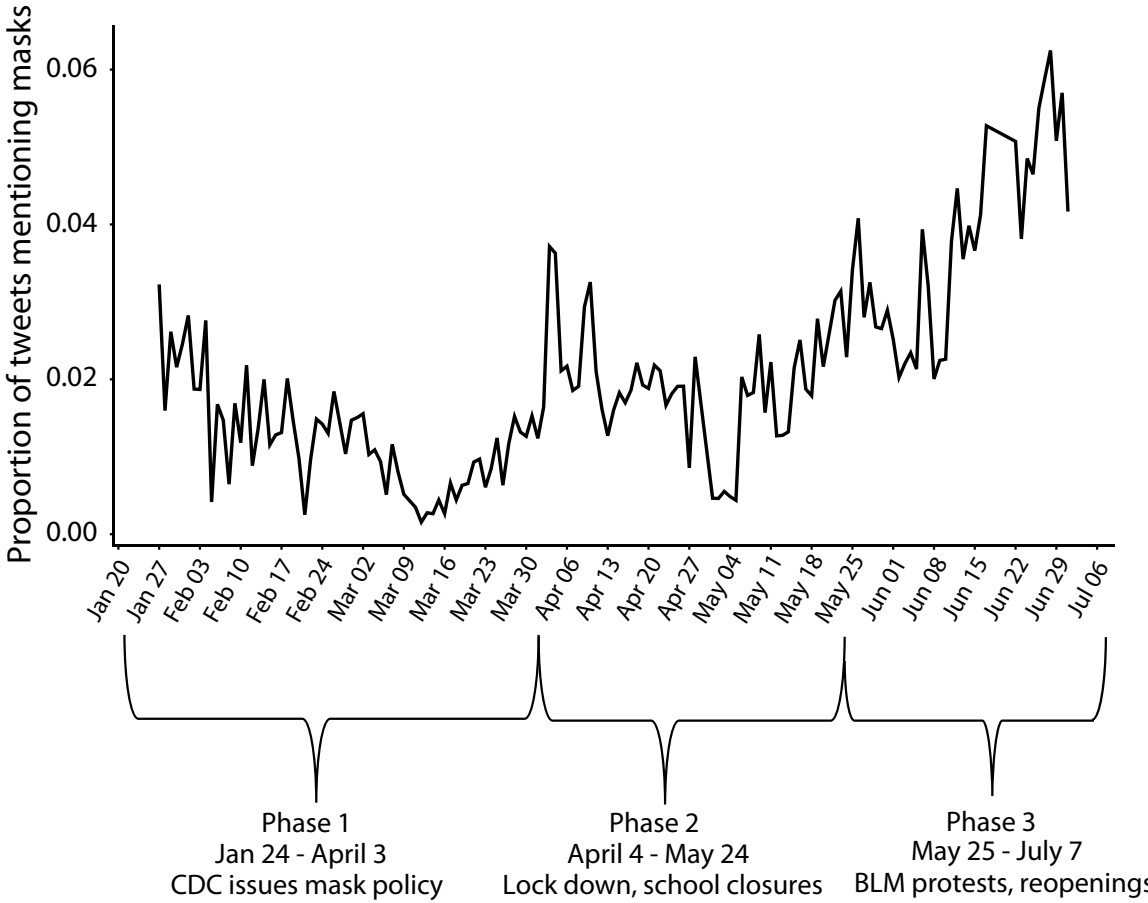

**Fig 1. Proportion of mask tweets (N = 7024) of 592,317 COVID-19 tweets between January 24 and July 7, 2020 across pandemic phases.**

Use of hashtags to amplify messages were used among 32.6% of twitter users (N = 2286/7004 tweets) with a total number of hashtags used (N = 5811/7004) reflecting 82.9%. Use of mentions to amplify or attenuate mask messages were used 39.6% (N = 2772/7004) with a total number of mentions (N = 4817/7004) reflecting 68.8%. Use of hyperlinks in mask tweets comprised 45.4% (N = 3182/7004).

We describe next how the six derived mask-wearing Twitter discussions were expressed and how tweets intensified and attenuated COVID-19 risk perception.

## COVID-19 severity perceptions

Sense making of the new COVID-19 health risk was expressed in part through COVID-19 severity perception. Mask tweets expressed the severity of the emerging pandemic in a range of ways, from discussions about emergent social behaviors (e.g., hoarding of masks) to sharing reports of hospitalizations, hotspots, and deaths to comparisons with known risks like flu or automobile accidents. Early mask tweets (January 30, 2020) expressed little fear or concern by

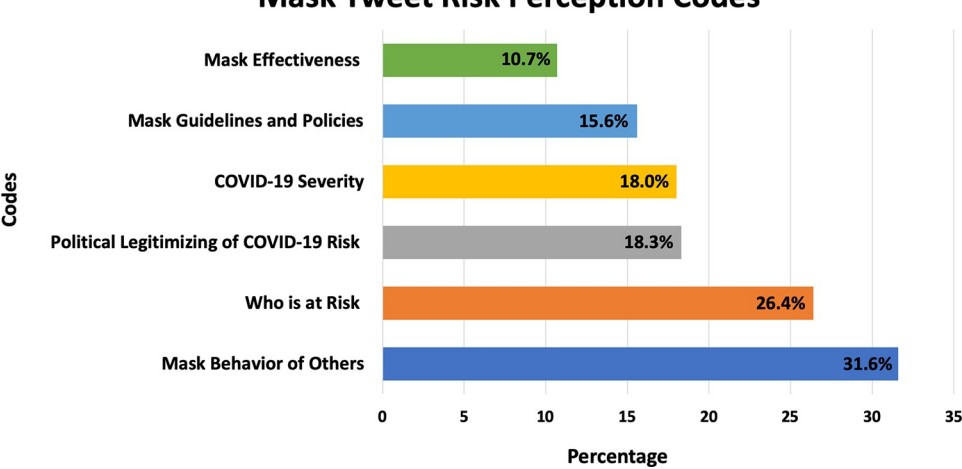

**Fig 2. Six derived mask tweet COVID-19 risk perception codes.** Each mask tweet could be coded for more than one risk perception category therefore, percentages do not add up to 100%. Mask effectiveness (N = 749/7004; 10.7%); Mask guidelines & policies (N = 1091/7004; 15.6%); COVID-19 severity (N = 1260/7004; 18.0%); Political legitimizing of COVID-19 risk N = 1281/7004; 18.3%); Who is at risk (N = 1850/7004; 26.4%); and Mask behavior of others (N = 2214/7004; 31.6%).

some in public settings, e.g., "the fear of the Wuhan virus is none on the Kriger train (northern California) today and no one is wearing a mask" to later in the pandemic, in June 2020, when tweets referred to second waves. Assessments of severity began with questioning and

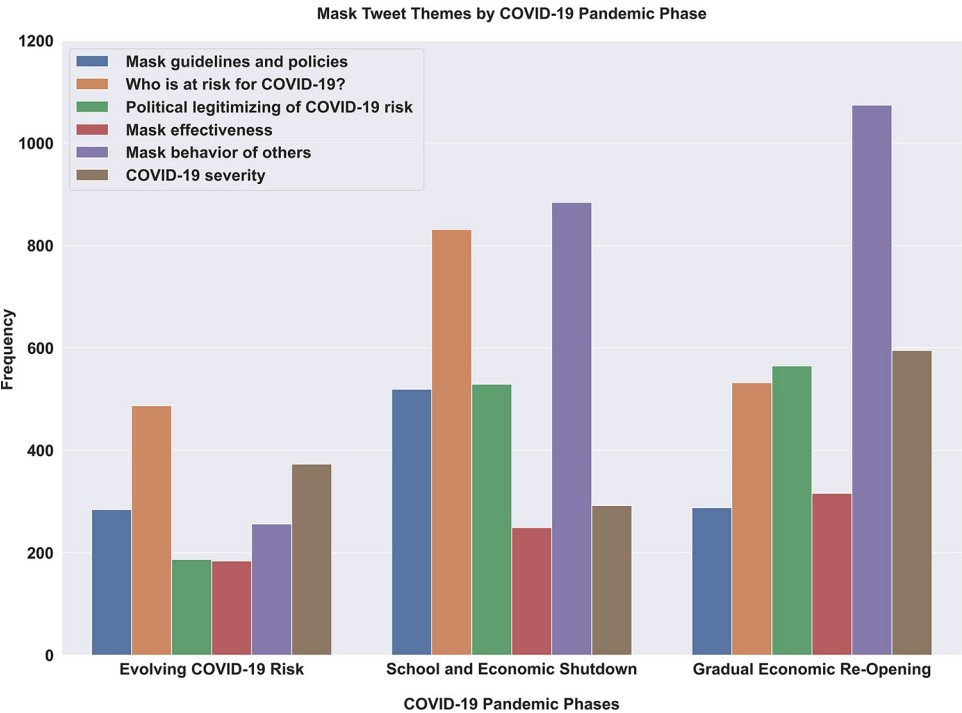

**Fig 3. Frequency of the six COVID-19 risk perception themes for the three major phases of the early pandemic response between January 24 and July 7, 2020.** The three phases are: Evolving COVID-19 risk (1/24/20–04/03/20), School and economic shutdown (04/04/20–05/25/20), and Gradual Economic Reopening (05/26/20–07/07/20).

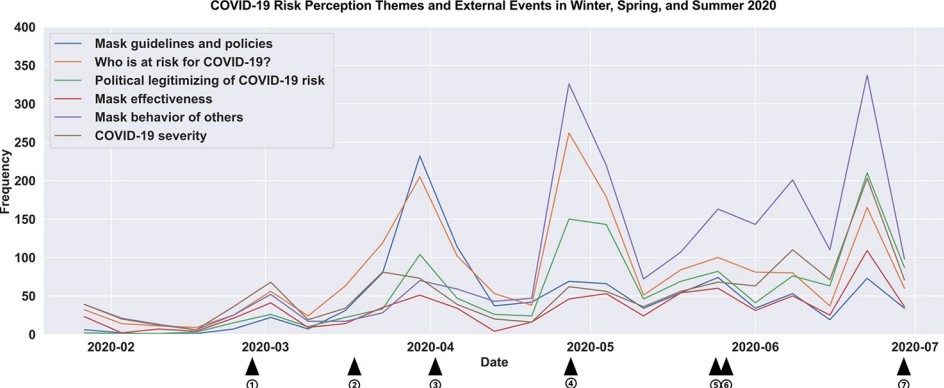

① 2/29/2020 U.S. Surgeon General Dr. Jerome Adams tweets that masks are not effective in protecting the general public. He emphasizes that masks should be reserved for healthcare providers.

② 3/19/2020 The U.S. State Department advises Americans to avoid traveling internationally and for those abroad to return to the U.S. immediately.

③ 4/3/2020 The White House and the CDC recommended Americans wear non-medical cloth face coverings in public. Trump emphasizes that wearing a mask is voluntary and said he will not be doing it.

④ 4/28/2020 Vice President Mike Pence toured the Mayo Clinic's coronavirus testing labs but courted controversy for ignoring the Minnesota hospital's rules that all occupants wear a mask.

⑤ 5/25/2020 Trump does not wear a mask during Memorial Day events. He later retweets a Fox News commentator criticizing Joe Biden for wearing a mask.

⑥ 5/27/2020 Dr. Fauci calls for Americans to wear a mask saying that while it's not 100% effective, it shows "respect for another person."

⑦ 6/29/2020 White House press secretary Kaleigh McEnany says that Trump insists mask wearing is a 'personal choice'.

**Fig 4. COVID-19 risk perception themes and external events in 2020.** Data points are aggregated by 7-day periods (weeks). For each month in the graph, there are 4 data points. Each data point represents the frequency for a certain label during the past 7-day period.

information seeking, e.g.: "Can we take a timeout and start gathering information on the Coronavirus? Where can you get masks? How to disinfect your work/home? What to teach your kids about other kids? CDC?" (February 24, 2020, Maryland).

Amplification of COVID-19 risk related to its perceived severity was expressed in a number of ways. Early pandemic phase mask tweets expressed severity by discussing mask wearing by the public as unusual and signaling alarm; rationing of mask supplies for short-staffed health care workers; hoarding and panic prescription filling; sharing media or public health reports of increases in coronavirus cases, hospitalizations, deaths, and hotspots; and mass production of masks signaling the magnitude of the evolving risk. Later pandemic phases expressed severity also by numbers and magnitude of harm, i.e., spreading cases, but also by overwhelmed hospital capacity, including intensive care unit bed capacity at maximum, increasing deaths, and hotspots.

In addition to tweet content amplifying risk perceptions, how tweets were shared also amplified risk perceptions. For example, tweet expressions used ALL CAPS to denote intensity; directed tweets at public officials through the use of mentions, e.g., @governors @POTUS, of those deemed responsible for (and failing) to manage the risk; used hashtags that expressed negative emotion, e.g., #scary; and shared photographs (e.g., empty grocery and supply shelves) and hyperlinks to news stories that amplified risk perception. The following exemplars illustrate the intensification of Covid-19 risk perception:

**Sharing reports of COVID-19 cases in March 2020.** Taking a plane back to Manhattan where the first case of coronavirus has just been confirmed, is triggering major anxiety. @Mabel_Syrup, can you remind us what face masks we should buy to protect ourselves? (March 2, 2020, FL).

**Short-staffed paramedics and emergency medical technicians.** FDNY short-staffed paramedics and EMTs expect to be overwhelmed by the coronavirus. N95 masks are being

rationed, exposed medics aren't being tested, and more then 150 FDNY members are now in quarantine https://t.co/5pYiScfMHf (March 20, 2020, NY).

**Mask wearing alarming.** "Not going to lie, people wearing masks kinda freaks me out! (May 3, 2020, MN).

**Sharing of case and death counts.** We just passed 10 MILLION Cases of #coronavirus The death toll is at 500,000. That's a staggering 5% mortality rate (not to mention the quality of life of many of the survivors) People opting out of masks imagine killing 1 of every 20 of your friends. 2 Kids per classroom. Dead https://t.co/X1mGsWkoSV (June 29, 2020, CA).

Attenuation of perceived COVID-19 risk severity was equally expressed. Risk expressions downplayed COVID-19 risk in the following ways: (a) comparisons with familiar risks like influenza, i.e., "the flu" or car accidents; (b) tweets from regional parts of the country that were not yet witnessing the spread of the COVID-19 virus like rural and Midwestern states; (c) tweets claiming COVID-19 risk was not real; (d) tweets expressing case numbers were exaggerated; and (e) tweets about continuing routine habits (e.g., continuing with air travel), as shown by these examples:

**Flu comparisons downplay COVID-19 risk.** The coronavirus is the same level virus as the flu strand lol. Y'all gotta chill. . .If you're using soap and water, wearing mask around sick people and in crowded areas you should be ok. Go on that trip. (March 3, 2020, TX).

**Rural states not yet experiencing COVID-19.** My state of Indiana had 54 common flu deaths, and zero COVID19 deaths. There was no statewide closing of bars and restaurants over the 54 common flu deaths, no buying up of toilet paper, surgical masks or hand sanitizer. It all needs to be put into perspective (March 16, 2020, IN).

**Reported exaggeration of hospitalizations.** Total current covid hospitalizations in Georgia are 783, that's 0.0007% of the population. Can we stop with the mask and the fear now? https://t.co/Ss57URVrEV (June 7, 2020, GA).

## Mask effectiveness

Debates about mask effectiveness implicitly and explicitly expressed COVID-19 risk perceptions. Mask effectiveness tweets disputed COVID-19 transmission routes, sharing "evidence" through news stories or academic articles and anecdotal hearsay from a range of "experts." Anecdotal evidence ranged from repeating what public health officials stated to sharing what "the neighbor who is a TSA employee and knows the latest updates" said to sharing evidence that masks work because people in close proximity did not get infected. Finally, personal opinions were commonly shared. Other mask effectiveness tweets discussed the size of virus particles, conflating the size of the SARS-CoV-2 virus with air pollution particulate matter or how masks could be altered (e.g., coating masks with salt to neutralize the virus). Mask effectiveness conversations, therefore, shared a mix of anecdotal to verifiable evidence shaping interpretations of COVID-19 risk. Example tweets illustrate the different "evidence sharing" and "expert sources" mentioned.

**Sharing expert source evidence about debate and transmission route.** LA County health officials say coronavirus is not a problem in SoCal; people shouldn't worry about it. They add that those masks people are wearing provide no protection against *any* respiratory illnesses. The key is thorough and frequent hand washing. #coronavirus (January 31, 2020, CA).

**Surface transmission emphasis.** Coming from a public health professional, using masks to prevent COVID-19 is NOT effective. You can still obtain the virus through touching. Our main culprit is our hands and the only effective prevention as of right now is hand washing and social distancing. #COVID2019 https://t.co/kjEGIXLaqR (March 17, 2020, NY).

**Sharing scientific evidence (academic) of mask effectiveness.** A new peer reviewed study found that public use of face masks prevented 66,000 coronavirus infections in NYC between April 6th and May 9th. The infection fatality rate for NYC is 1.4%. In one month, in one city, face masks prevented almost 1000 deaths. https://t.co/ZiPFpxziRe (June 30, 2020, MN).

Other mask effectiveness discussions expressed uncertainty about the effectiveness of different mask types and who masks protect. These expressions were often accompanied by explicit calls to wear a mask and expressions of lost credibility and trust in public health institutions due to discrepant messaging. At the same time, other mask tweets doubting their effectiveness made judgements about individuals wearing masks (e.g., stating that people wearing masks look like fools).

## Who is at risk?

Tweets about who is at risk included sharing (a) media reports about who is at risk of COVID-19 in the United States; (b) personal experiences of getting sick; (c) family and friend networks experiencing COVID-19 illness or not; and (d) discussions about subgroups at risk of acquiring COVID-19 or blamed for spreading COVID-19 risk. Descriptions of personal experiences with COVID-19 amplified risk perceptions when the experiences involved describing severe or adverse COVID-19 symptoms, whereas asymptomatic COVID-19 experiences attenuated risk perceptions. Sharing of adverse personal experiences prompted users to endorse mask wearing to mitigate risk.

**Personal network COVID-19 experience tweet.** My BFF's dad died last night. Alone, without his family. Because of COVID19 care homes are locked down. They were only allowed to see him once in the past 6 wks and only for a few minutes. Let that sink in before you go out without a mask to celebrate Memorial Day (May 25, 2020, CA).

**Secondhand COVID-19 experience tweet.** I was pleased to see my Uber driver wearing a mask tonight—then he told me his mother had just died of #COVID19—I just listened while he told me the whole story of her life—I'm just numb. I was coming back from the White House—it was the most real thing I had heard all day (March 26, 2020, Washington D.C.).

At the same time, although learning that COVID-19 could be deadly, others shared stories that COVID-19 risk did not result in serious illness.

**Personal network not experiencing severe disease.** Since COVID19 started I have been out literally EVERY WEEKEND around crowds of people, no mask on. . .yet I have never experienced any symptoms. No cough, fever, nothing. . .I'm healthy as a horse. Why is that? (June 9, 2020, NC).

Subgroups mentioned included travelers (both for spreading the virus and putting themselves at risk); older adults and those in long-term care facilities (being vulnerable); doctors, nurses, paramedics, first responders (i.e., health care workers) being vulnerable; those who are immunocompromised and fighting cancer; and essential workers like plant workers. In later phases of the pandemic, tweets described additional subgroups like young adults, black lives matter (BLM) protestors, travelers, and children as responsible for spreading the coronavirus and putting others at risk when not wearing masks. Stories of subgroups involved both subgroups vulnerable to disease risk and being blamed for putting others at risk.

**Health care workers at risk due to absence of workplace protocols.** I am very, very worried about #COVID19—I work in a large ER. In my opinion we should all be wearing N95 masks. We aren't even being encouraged to wear masks because of the shortage of masks. N95 masks are on a need to have basis. . .and I'm not sure who makes that decision. (March 10, 2020, GA).

**Young people being blamed for spreading COVID-19.**   @GregAbott_TX Let me get this straight, you're blaming young people for Texas' RAPID covid spike, when you're the one who opened the state way too soon, downplayed the need to wear masks, & have taken away the ability for mayors to enforce masks & distancing guidelines #f*ckyou. (June 16, 2020, TX).

## Mask guidelines and policies

Mask tweets expressed the contradictory public health guidelines issued by the CDC and public health officials and discrepant messaging and behavior by government officials.

**Discrepant mask guidelines.**   Seems they want everyone to get sick. For three months they tell everyone not to wear a mask while the airborne Coronavirus spreads uncontrollably. Then when they finally tell people to wear a mask they recommend using a method that is least effective. https://t.co/yXdJ5rjf7d https://t.co/UFNycLreAL (April 4, 2020, DC).

Workplace protocol changes in select work environments signaled the prioritization of COVID-19 risk. Tweets described workplace mask mandates for occupations from construction and subway workers to police and restaurant workers.

**Workplace policies recognizing risk (subway workers).**   @NYCTSubway workers are given one mask and one pair of gloves for an entire week while continuing to work. I had no idea that's all the PPE they get. Ironically, the majority of these subway workers are, you guessed it, black and latino. #ShutTheNYCSubwayDown https://t.co/8x4VzufIYs (April 18, 2020, NY).

Tweets ranged from reactions to mandatory mask wearing in some workplace settings while not enough enforcement in other settings and feeling at risk because others at work were not wearing masks.

## Political legitimizing of risk

In this theme, mask tweets centered on (a) reactions to government (White House) messaging; (b) discussions about the lack of mask role-modeling by government authorities; and (c) state policy messaging by governors, mayors, and state and county public health agencies. Mask tweets referenced, through mentions, government and public health authorities, ranging from the former president, federal administrators, and the World Health Organization to governors and local public health agencies (see Table 1). Tweets across states and time reflected the evolving policies and conflicting messages shaping risk perception. Tweets mirrored in many cases the ongoing power struggle between the White House, which initially downplayed COVID-19 risk, and the CDC. Mask tweets also commented on former President Trump not role-modeling mask wearing and not implementing a federal plan to mitigate COVID-19 risk.

**Former President Trump and Vice President Mike Pence not role-modeling mask wearing.**   Good lord—all the president and VP have to do is wear masks and encourage others to do so as well. But that would mean they are leaders. https://t.co/2XiBjHu9Lx
(May 2, 2020, GA).

**Former President Trump downplaying COVID-19 risk.**   trump on @espn yesterday said that he wants sports fans to go back to normal, no masks and on top of each other. Is this the official @WhiteHouse advise? Does @HHSGov and @CDCgov have any opinion on this? @maddow @NicolleDWallace (May 18, 2020, NV).

**Growing distrust in government authority.**   I believe this doctor over the person in the Oval Office for COVID19 guidance!! Thank you Dr. Fauci for your service. @DrAnthony Fauci says he wears a mask to be a symbol of what "you should be doing https://t.co/h2NHL0hsVd (May 27, 2020, TN).

**Table 1. Government, health, and other institutions mentioned (@) in mask tweets.**

| International public health organizations | World Health Organization |
|---|---|
| Government institutions | Centers for Disease Control and Prevention |
| | National Institute of Health |
| | Federal Emergency Management Agency |
| | US Department of Health and Human Services |
| | US Department of State |
| | Federal Bureau of Investigation |
| | Central Intelligence Agency |
| | Department of Justice |
| | Former President Trump |
| State representatives | Katie Porter (CA) |
| | Jimmy Gomez (CA) |
| State public health departments | California (first state to implement shelter-in-place policy) |
| | Michigan |
| | Ohio |
| County public health officials | Dr. Barbara Ferrer (Los Angeles County public health director) |
| Governors | Gavin Newsom (CA) |
| | Andrew Cuomo (NY) |
| Mayors | Eric Garcetti (Los Angeles, CA) |
| U.S. public health institutions | American Public Health Association |
| Academic institutions | Johns Hopkins University |
| | Georgia Southern University |
| | UCLA Health |
| | University of Birmingham |
| Media sources | Rachel Maddow (CNN) |
| | NPR (local) |
| | The New York Times |
| | John Oliver (comedian and TV host) |
| | Fareed Zakaria |
| Physicians | Dr. Anthony Fauci |
| | Dr. Amy Acton (director of Ohio Department of Health who stepped down after receiving threats for mask policies) |
| | Dr. Sanjay Gupta |
| Private individuals | Bill Gates |
| | Former President Barack Obama |
| 2020 presidential candidates | Bernie Sanders |
| | Joe Biden |
| | Elizabeth Warren |
| | Mike Bloomberg |
| Economists | Scott Minerd |
| | Mohamed El-Erian |
| | Noriel Roubini |
| Apps | Weather Channel (coronavirus tracking) |

**Power struggles between government and public health authorities.** Watching the coronavirus WH task force briefing and you can see it behind his mask but Dr. Fauci is FED UP with the government (June 26, 2020, TX).

**Partisan comments about mask wearing emerging.** Republicans, do any of you support wearing masks to reduce the spread of covid? It seems like your head honcho has politicized something that's just common sense responsibility (June 27, 2020, NC).

Tweets discussed politics as it relates to risk for Americans, with discussions linking political ideologies and personal mask-wearing behavior.

**Political ideology shaping response to public health recommendations.** Wearing masks should not be a political issue. #COVID19 does not discriminate based on political ideology. Wearing masks work and the work we have done collectively has helped save lives (May 18, 2020, TX).

**Partisan reactions to mask wearing.** COVID became overplayed BS those liberal people who get mad for others not wearing masks r the same ones ok with mass protesters out an about not wearing masks. . .https://t.co/GIsrxHCP7Z (June 4, 2020, AZ).

Tweets covered messaging, policies, and handling of the crisis by American political figures including governors, mayors, and public health officials. Tweets also discussed mask mandates and observations of mask-wearing behavior by politicians at any level.

**Mentions of governors in mask-wearing comments.** @GovMikeDeWine [Ohio] CDC recommends healthy people should not wear masks. If you would like to continue to wear a mask please move to China. This is the United States of America and we should not be forced to wear masks (June 14, 2020, OH).

## Mask behavior of others

Composing the largest proportion of mask tweets, the theme of "mask behavior of others" described mask behavior in the context of daily life, including travel or commuting to work, social settings like restaurants and shopping, and people attending music concerts (see Table 2 for locations of mask-wearing observations). Twitter discussions changed over time, with early tweets describing alarm at observing mask wearing (January and February 2020), when mask wearing was not yet normative in the United States, to later pandemic phases (April to June 2020), when tweets expressed dismay at others not wearing masks after it was recommended

**Table 2. Locations of mask-wearing observations signaling risk in public settings, 2020.**

| |
|---|
| Airports (passengers, airport employees) |
| Subway, trains, light rail, buses |
| Cruise ships |
| Entertainment or recreation (theme parks) |
| Pharmacies |
| Grocery stores |
| Hardware stores |
| Restaurants |
| Bookstores |
| Beauty and nail shops |
| Strip clubs |
| Schools |
| Nursing homes |
| Sports venues (baseball fields) |
| Hospitals |
| Manufacturing plants |
| Meatpacking plants |
| Retailers |

for the general public to prevent the spread. Early observations (February 6, 2020) described the varying levels of risk perception tweeting, e.g., about one passenger on a commuter train who wore a mask next to another person showing little concern and making a peanut butter and jelly sandwich. Later phases (May 15, 2020) illustrated how "social cues of others" contributed to shaping interpretations of risk, e.g., "Bruh if covid-19 is so mf serious why aren't they wearing masks?" (May 15, 2020, NC) and "I went to the dollar store and a bunch of ppl weren't wearing masks and they were just chilling in the store, this don't make sense" (May 2, 2020, TX).

**Mask wearing in public alarming people in early pandemic phase.** Why are some people wearing masks on Philadelphia trains? It is freaking me out. #Philadelphia #coronavirus (February 3, 2020, PA).

**Observing others not wearing masks.** @YNB There were so many people out shopping without masks yesterday that I predict our second awful wave of COVID19 will hit by July 4. There's where your "independence" gets you! (May 24, 2020, CA).

Tweets about observations of others not wearing masks were often accompanied by calls to wear masks using hashtags and altruistic appeals ("take care of your neighbors and wear your mask tomorrow, wearing a mask says I respect you and the wellness of those you love and care for #BeSafeKentucky") or guilt appeals ("Arkansas high school party triggers outbreak, if you are not wearing a mask you are putting everyone at risk of getting sick, #selfish, #mask4all, #wearamask"). Over time, as COVID-19 risk became more evident across the country by the rising number of deaths and geographic spread, the emotional tone of tweets expressed increasing anger by some, and the nature of hashtags and use of expletives reflected this: "It's crazy that over 72,000 people have died in a couple of months from COVID-19 and you can't wear a f*ing mask #selfish" (May, 2020, OK). Even the governor of California, Gavin Newsom, shared an expletive hashtag on June 22, 2020: "#wearadamnmask" (see Table 3 for how mask hashtags changed during the first 5 months of the pandemic). Observation of others also included improper mask wearing: "I see so many people wearing masks but not over their nose, WTH?" (June 23, 2020, CA).

Mask tweets about the mask behaviors of others also included role-modeling mask wearing. Explicit mentions of role modeling referenced journalists and celebrities, e.g., "good to see Sid Hartman sporting a @twins #Minneapolis mask" (April 8, 2020 MN) whereas other high-

**Table 3. Mask hashtags.**

| February 17 | #mask |
| --- | --- |
| February 24 | #N95mask |
| March 2 | #facemask |
| March 20 | #PPE |
| March 21 | #maskshortage |
| March 23 | #maskforall |
| April 4 | #wearamask |
| April 4 | #MasksSaveLives |
| April 30 | #newnormal |
| May 11 | #maskup"City" (PHL) (HOUSTON) |
| June 1 | #nomask |
| June 15 | #maskupAZ |
| June 22 | #wearadamnmask |
| June 26 | #realmenwearmasks |
| July 10 | #heroeswearmasks |

profile political pundits were observed not role modeling: "Coronavirus stricken George Stephanopoulos ignores mask mandate during Hamptons stroll" (April 21, 2020, NY).

Observations of mask wearing becoming normative over time were expressed in a variety of ways: "We're all just doing this? All of us wearing masks and acting like this is completely normal? Cool." (May 1, 2020, IA); discussions of mask wearing as a daily ritual when leaving home, like grabbing keys, wallet, phone, and a mask; sharing internal thoughts, e.g., "I never thought I'd be comparing people's masks as I grocery shop" (April 9, 2020, VT), and using hashtags like "#NewNormal." These tweets reflected an increasing acceptance of Covid-19 risk as part of daily life.

Expressions of tweets ranged from commentary to judgements, calls to wear or not wear masks, and hypothetical discussions about others' mask-wearing behavior and potential consequences (e.g., putting others at risk). The tweets of medical, scientific, or public health figures (e.g., Dr. Anthony Fauci), journalists, celebrities, or those in a personal network either modeling or observing behavior of others expressed reactions to mask wearing becoming normative over time, e.g., "So now we need our hat, keys, coat and mask to leave the house? #CoronavirusOutbreak" (April 11, 2020, NC). Other examples illustrate how mask wearing was initially not normative and provoked certain reactions to mask wearing becoming normative and eliciting different reactions later in the pandemic:

**Sharing experience of wearing mask when no one else was.**   Now the CDC recommends everyone to wear masks and if you don't have a mask, wear a scarf around your mouth and nose. Let me tell you, I went to CVS dressed like that and people looked at me like I was a criminal. Hard pass. (April 4, 2020, PA).

**Behavior of others.**   This is not acceptable! #AmericanAirlines @AmericanAir has my husband and daughter inside the plane for almost 3 hours @LGAairport due to mechanical issues. There are people with NO masks inside the plane #pandemic #flight1788 #CoronavirusOutbreak (June 7, 2020, NY).

## Discussion

This empirical study described through Twitter the public response to mask wearing and perceived COVID-19 risk during a critical time in the evolution of the SARS-CoV-2 outbreak between January and July 2020. We investigated how mask tweets expressed COVID-19 risk perception, focusing on the antecedents that shaped risk perception and how they changed dynamically over time. We did this through the perspective of citizens who actively participated in shaping risk perceptions as "individual amplification or attenuation stations," by responding to, commenting on, and actively sharing risk messages with select networks, as documented on social media. Psychology and communication research have shown that perceived not objective risk motivates behavior [12, 13, 35, 36]. For this reason, it is important to understand the how COVID-19 risk perception was diffused, amplified and attenuated as the pandemic crisis unfolded. As Twitter users shared mask tweets with their networks, and included hyperlinks, hashtags and or mentions these communication acts (as well as what content was passed on) shaped COVID-19 risk perceptions, which and may or may not have had consequences for mask-wearing.

We discuss implications of the six derived themes reflecting social media conversations about COVID-19 risk, how risk was expressed, and how it changed over time from risk severity perceptions to mask effectiveness debates, questions and discussions about who is at risk, debates about mask guidelines and policies, the political legitimizing of risk by government officials (or lack thereof), and the role of mask behavior of others in shaping risk response. The themes reflect subjective rather than objective constructions of risk.

We examined the flurry of Twitter discussions of COVID-19 risk and related communication as it unfolded in real time, shaping risk perceptions, responses, and consequent risk mitigation mask-wearing behaviors. Our analysis of mask tweets during 2020 describe a history of the American societal reaction to COVID-19 risk and show that risk perceptions were expressed in a much more expanded range than "objective, rationale" risk. While mask effectiveness tweets debated transmission routes, other mask tweets particularly "behavior of others" attest to the social experiences that contributed to constructions of COVID-19 risk that in turn, motivate behavior. Twitter users took advantage of the affordances of the social media platform to document, engage, amplify, and attenuate COVID-19 risk perceptions, including making sense of the unfamiliar risk. Mask tweets reflected the mediated and shared experiences of COVID-19, passing on information through weak and strong network ties, adding hashtags and hyperlinks to amplify or attenuate messages, and using mentions to draw the attention of various government authorities and stakeholders. Tweets expressed both the social experiences and the isolating and threatening experiences that COVID-19 posed as parts of the country experienced lockdowns, shelter-in-place orders, and school closures while the virus spread. Responses to mask wearing evolved from initial signals of alarm to calls for all to wear masks as desperate pleas to mitigate the very real and collective risk to society increased.

Severity of COVID-19 risk was expressed in various ways because the risk was initially unclear. COVID-19 severity perception was intensified by descriptions of increases in hospitalizations, deaths, intensive care unit beds at capacity, and geographic hotspot discussions. However, it was also downplayed by comparisons with the flu, rural states not yet experiencing COVID-19, and reports of exaggerations or that COVID-19 is a hoax. Severity assessment as an aspect of risk perception has been recognized in multiple health behavior change theories [37, 38]. Amplification occurred in risk expressions as feelings, e.g., through expressions of dread and negative emotions (anxiety and fear in both content and hashtags, e.g., #scary), discussions about hoarding and rationing of masks, and observations about mask wearing signaling alarm while attenuation occurred in expressions of ambivalence about mask wearing.

Mask effectiveness debates peaked end of March through early April 2020, when the CDC issued guidelines for the asymptomatic general public to wear masks in public settings [39]. Mask effectiveness debates heightened the uncertainty surrounding the science and transmission of SARS-CoV-2 (contact, droplet, airborne), casting doubt on recommendations. Early in the pandemic, surface transmission was emphasized by public health and government officials, with an emphasis on hand washing and cleaning surfaces [8]. Public health officials did not mandate mask wearing early and even explicitly recommended not wearing masks for the general asymptomatic public [39]. Mask effectiveness tweets explicitly and implicitly shared the relatively more "objective" evidentiary appeals that signaled risk and contributed to interpretations of risk (implicitly related to transmission routes).

Debates brought attention to COVID-19 risk and heightened uncertainty around whether the recommended risk mitigation behavior was warranted, with several studies having been reported since to justify the policy of mask wearing for SARS-CoV-2 transmission control [5, 8, 40–42]. Discussions about risk also centered on not only transmission routes, but also who is vulnerable.

A continuous assessment of the unfamiliar and evolving COVID-19 risk was expressed in tweets about who is at risk to assess risk susceptibility in the sense-making process. Discussions both amplified and attenuated risk. People sharing experiences of family members or the users themselves getting very sick or dying amplified risk perceptions, whereas others sharing that they did not get sick despite being in public settings or large gatherings attenuated risk perceptions. Tweets about who is at risk represented the second largest volume of tweets after those

discussing the behavior of others. In these tweets, individuals learned about the unfamiliar COVID-19 risk through the mediated experiences of others in lieu of direct experience.

Risk perception literature refers to the learning of risk through mediated, shared experience of others as an availability heuristic, wherein vivid examples are judged as reflecting higher risk [43, 44] and a scenario is activated when people are confronted with a new and complex risk with which they have limited personal experience [44–46]. Personal stories served to make salient and amplify the invisible and psychologically distant risk for some. In construal level theory [47], the perception of risk is in part shaped by subjective and psychological proximity, with more direct exposure and experience to risk resulting in greater risk perceptions if the hazard (disease, in this case) is observed.

Discussion of subgroups and who is vulnerable signaled risk perception as group identity (e.g., travelers, health care workers, older adults, essential workers, children, young adults, men). Children were initially deemed at low risk and spared severe disease. Discussions initially centered on health care workers and those living in long-term care facilities, followed by essential workers. Individuals discussed changing workplace protocols or alternatively, no change in workplace setting mask-wearing protocols and policies. Understanding how social identity shapes risk perceptions has played a significant role in understanding and effectively approaching behavior change across behaviors—for example, smoking cessation [48–52]. Twitter discussions suggest that social and political identities shaped COVID-19 risk attitudes and consequent response to risk mitigation recommendations [53].

Discrepant federal guidelines represented another way in which the public discussed and made sense of COVID-19 risk. The public takes its cues about which risks to prioritize, in part, from government and authority institutions. Risk governance theory recognizes the role of risk regulations by social institutions drawing boundaries around what is considered acceptable and not acceptable risk [54–56]. Risk governance theory recognizes that the public is not homogeneous and that different subgroups will perceive risk differently [56, 57]. For this reason, identifying social determinants of risk perceptions becomes increasingly important in explaining risk response.

Discussions about the CDC mask guidelines peaked in early April, when the CDC issued its recommendation for the asymptomatic public to wear masks in public settings (April 3, 2020). Mask tweets expressed the discrepancies in CDC guidelines, reacting to the about-face and months-late recommendations. Public response to the CDC changing its official guidelines signaled that citizens felt the risk was more widespread than publicly acknowledged by government authorities, resulting in lost credibility and trust in the government institution. The CDC director, Robert R. Redfield, stated on February 27, 2020, "There is no role for these masks in the community," adding, "These masks need to be prioritized for health care professionals that as part of their job are taking care of individuals." Also, U.S. Surgeon General Dr. Adam Jerome tweeted on February 29 to not purchase masks: "Seriously people—STOP BUYING MASKS! They are NOT effective in preventing general public from catching #Coronavirus, but if healthcare providers can't get them to care for sick patients, it puts them and our communities at risk! https://t.co/UxZRwxxKL9."". When the CDC released guidelines to wear masks in public on April 3, 2020, Twitter reactions referenced the discrepant messaging and spiked [39].

Government institutional messaging from not only federal public health institutions like the CDC but also the U.S. government contributed to COVID-19 risk perception and mask-wearing attitudes. Tweets by former President Trump, former Vice President Pence, and various White House officials and governors were discussed on Twitter in relation to masks. Mask tweets amplified COVID-19 risk perception through explicit mentions of government officials, especially governors. When former President Trump did not wear a mask and played down

COVID-19 risks, these messages and the absence of mask role modeling resulted in politicization of the handling and management of COVID-19 risk mitigation. Risk governance theory recognizes how risk response is shaped by social interactions among actors including government, whose messaging is no longer a centrally controlled creation [54].

Tweets reflected a range of responses referencing the politicization of mask wearing. Mask wearing being ascribed many meanings and deeply connected to social and cultural practices has also been observed since from interviews with from health experts across the globe [58]. Mask wearing was consequently discussed on Twitter not only as protecting against COVID-19 as a collective risk, but also as a form of government control. Mask tweets mentioning government authorities and politicians attest to the significant role government officials play in acknowledging, legitimizing, and prioritizing competing risks for the public. Politicization of mask wearing in the United States resulted in attenuation of risk perceptions for some and intensification for others as mistrust grew. Politicization began with former President Trump downplaying COVID-19 risk, not acknowledging its seriousness, and not role-modeling mask wearing. By not wearing a mask, the president's actions symbolically signaled Covid-19 as lower risk priority.

Although government mask guidelines typically serve to sanction and legitimize risk, politicization in the case of COVID-19 under the Trump administration undermined the credibility that institutional signaling plays in legitimizing a public health risk as a prioritized threat [59]. As sociologist Ulrich Beck argued, we live in a risk society with everyday competing risks, and in this capacity, institutional messaging serves to legitimize risks that warrant prioritization [60]. With the science of COVID-19 risk evolving and not clearly communicated, the public expressed and directed questions about whether mitigation measures such as mask wearing were necessary at political, government and public health officials amplifying risk and confusion.

Tweets about the mask behavior of others comprised the largest category of mask conversations related to COVID-19 risk perceptions on Twitter. Tweets observing mask wearing in public settings during travel (air, cruise ship, or commuting on public transport) in early 2020 signaled and intensified risk perceptions by visibly linking risk typically associated with hospital settings to public settings and social activities (e.g., travel, grocery store visit) whereas later phases reflected to some degree normalizing mask wearing. This change in mask wearing from a medical to public setting reflected a dramatic shift in the public's psychological bearing and grasping of COVID-19 risk and is in part reflected in mask wearing reactions. This experience has in fact also been observed in other countries across the globe despite other customs [58].

The public and collective nature of COVID-19 risk led to social relational norms shaping risk perception as the pandemic's toll on society grew over time. Social norms shaping risk perception and consequent behaviors have been observed across health behaviors [61–63]. Over time, mask wearing shifted from being a personal, individual behavior change to reflecting social responsibility at the collective level. Over a third of mask conversations on Twitter reflected observations of others wearing or not wearing masks over time. Former President Trump not serving as a role model with respect to mask wearing led to partisanship regarding the public health recommendation to wear masks and resulted in some citizens questioning public health guidelines and even whether COVID-19 risk was real [9]. Furthermore, the meaning ascribed to mask wearing in the politicized context changed over time, with mask wearing signaling social identity for some, whereas for others, it signaled the collective effort to mitigate spread of risk (after April 3, 2020).

Changes in workplace norms and policies were also discussed in mask tweets and shaped risk perception by signaling, through workplace protocols, recognition of the public health risk of coronavirus as real, serious, and to be prioritized. This was observed by tweets of

employees, especially essential workers like Uber drivers, restaurant workers, office workers, subway workers, construction workers, aviation maintenance workers, and plant workers. Risk signals from workplace policies play an important role in sanctioning a risk as legitimate and consequently, contributing to interpretations of risk.

### SARF insights: Social media mechanisms of amplification and attenuation

SARF focuses on receiver reactions to risk messages to gain insight and explain divergence in meanings that individuals ascribe to risk. This study contributes to the risk perception literature by identifying how the public discussed COVID-19 risk, i.e., citizens who shared unsolicited and authentic tweets reflecting their thoughts, and uncovering how social experiences of risk contributed to shaping risk perceptions. Many tweets reflected responses and reactions to the social behavior of others and political officials, interacting to shape dynamic and evolving public COVID-19 risk perceptions and negotiating and contributing to drawing boundaries and rules for acceptable behavior as it related to the novel COVID-19 risk. Amplification (i.e., intensification and attenuation) occurred simultaneously on Twitter as citizens exchanged risk information explicitly and implicitly.

Mechanisms of information transmission on Twitter that contributed to amplification and attenuation of COVID-19 risk in mask tweets occurred both through structural affordances of the microblogging platform (mentions, retweets, hyperlinks, images, videos) and also through volume and the way content was shared. Discussion about ambiguity in guidelines, disputes about the science of COVID-19 transmission, power struggles between government and public health officials, politicization, and social identity amplified and shaped risk perception (e.g., with mask wearing taking on new meaning for some). Debates between experts and government officials functioned to heighten uncertainty about the severity and prioritization of COVID-19 risk, the reliability of recommendations, and the trustworthiness of basic facts downplaying risk perceptions. Debates likely also increased public doubt about the state of knowledge and mask recommendations. Dramatization by former President Trump refusing to role-model mask wearing led to both heightened and attenuated risk perceptions. In contrast to increased media attention about the 2014–2016 Ebola virus outbreak [64] that reflected a negligible risk to U.S. citizens, risk from COVID-19 was downplayed by the government when, in fact, the real risk to the U.S. public steadily increased and was deadly, killing more than half a million people [2].

Our study describes six ways in which COVID-19 risk was discussed in mask tweets on social media, a platform that functions as a risk amplification and attenuation station through the nature of the content but also through mentions to government officials, journalists, and politician stakeholders (e.g., @POTUS, @Trump, @governor) [65]. SARF contends that social amplification of risk occurs at two stages: in the initial transmission of information about a risk (we examined the social media context) and in the response mechanisms of society (behavioral, economic, and symbolic impacts). Amplification and attenuation occurred simultaneously on Twitter, attesting to the multiple ways in which information flow in this environment functions in complex ways. User engagement in social media contexts is uncontrolled and can thereby complicate dissemination of official public health messaging.

### Implications for public health messaging

This study makes salient that citizens ascribed many meanings to mask wearing and COVID-19 risk. Not only were many meanings ascribed, but also risk perceptions were shaped by many levels of influence from relational aspects such as behavior of others to workplace policies and government guidelines. Mask wearing was interpreted not only as a risk mitigation

measure as it was intended, but also as symbolic of political ideology, government control for some–suggesting evidence for social constructions of COVID-19 risk.

The simultaneous amplification and attenuation of COVID-19 risk perception in mask tweets complicates the potential for effective public health messaging. Tailored messaging delivered to subgroups is likely warranted given the many meanings ascribed to mask wearing. A social relational approach is indicated based on mask behavior of others playing a role in risk perceptions. For example, to reach young adults, the largest unvaccinated group to date, public health messaging may be more effective by tapping into social identity and social relational norms given the role of mask behavior of others and in-group identity having influence. Collective responsibility messaging delivered by in-group members may be needed to effectively reach a particularly subgroup whether young adults, a particular subgroup who identifies with a race/ethnicity, or political affiliation.

Results indicate COVID-19 risk perception is shaped by institutional and guideline and policy level messaging. These results suggest that another public health messaging approach emphasizing structural determinants of risk that shifts messaging away from individual responsibility may have better success of acceptance given the mixed messaging occurring at the individual level. Public health messaging emphasizing workplace policies setting the norms for masking may have the best chance for success. This has been observed in the effects of policy level masking [66].

Given that the American public turns to social media platforms for crisis risk information and response, investment in monitoring social media sites as amplification and attenuation stations for public health risk is warranted. Public health officials will need to invest in building sites of influence and trust on social media platforms like Twitter. Rapidly relaying credible, accurate and timely information during disasters is increasingly expected [24]. Building trust and credibility on social media offers an opportunity to build public health influence to increase adherence with recommended behaviors that mitigate real risks. The United States was one of the few countries where mask wearing became politicized and controversial early in the pandemic [9].

## Limitations

Although the study provides theoretical and practical implications for future research, it is necessary to note some limitations. First, the study was based on a random ~1% sample of coronavirus and subsequent mask keyword-related tweets to examine COVID-19 risk perceptions. Different Twitter data-sampling strategies have shown that a random sample detected the same number of themes as a topic sample, suggesting that it was useful to qualitatively assess frequencies [67]. A second limitation involves the biases inherent in the data source of Twitter users, who represent citizens many of whom are young adults (ages 18 to 29 and comprise the largest group at 42%), more likely to identify as Democrat, to be highly educated, have higher incomes, slightly more likely to be men (62%), and more likely to be concentrated in urban and suburban geolocations [68–70]. Twitter users reflect a range of stakeholders, including media organizations. Therefore, we filtered out the top 100 media organizations from the final mask tweet sample to reflect predominantly citizens tweeting about masks and mask behavior. We also manually read and analyzed the 7,000-plus mask tweets to confirm the tweets were sensical and not generated by bots. A third limitation is that Twitter data covered the first 5 months of 2020 pandemic and may not generalize to Twitter discussions about later points in the pandemic after July 2020. Having said this, Twitter volume related to COVID-19 peaked during the first 5 months, representing a critical time period in which to analyze key conversational themes emerging about COVID-19 risk. Results may be generalizable to early critical

stages of a public health crisis. Future research investigating how structural social media message attributes (e.g., hyperlinks, images, videos, mentions, retweets) amplify and attenuate risk perceptions, in addition to a content will contribute to elaborating on the dynamic communication process shaping risk perception on social media. Retweets were not available in our dataset given Twitter policy at the time of data collection. We did capture the proportion of tweets using hyperlinks, mentions, hashtags and questions to begin to understand the use of social media structural communication strategies in addition to content.

## Conclusions

As social media grows in its function as a mainstream news platform, especially for accessing public health information during disasters and pandemics, public health and government officials will benefit from monitoring social media contexts that play a role in shaping citizen risk response. With Twitter being the most popular social network for news consumptions and having 69 million monthly active users in the US alone [70], it serves as a useful vehicle for learning about real-time public reactions to public health crises. Intensification and attenuation of COVID-19 risk communication on Twitter unfolded in unique, rapid, and broad ways during the first 5 months of the COVID-19 pandemic, contributing to diverging constructions of COVID-19 risk and consequently, divergent adherence to the recommended risk mitigation behavior of mask wearing. Individuals communicated their personal experiences, and real-time reactions to the emerging COVID-19 risk, sharing not only content, but directing reactions to the emerging risk, behavior of others and response to changing guidelines with the use of mentions, videos (both of observation of others and personal videos) as collective pleas were made to wear masks, as well as the emergence of mask hashtags about shortage of personal protective equipment (PPE) for healthcare providers to collective risk appeals with hashtags about masking for all or mask up "a city" and making appeals to social identities with #realmenwearmasks.

Results can be generalized to early critical stages of public health crises regarding factors that may influence risk perception. Particularly under conditions of high uncertainty, and ambiguity, normalizing expected risk mitigation recommendations (behavior of others) plays a critical role. Attention to how a public health threat is communicated with respect to signals of risk severity, who is at risk and group identity, behavior of others (e.g., norms, workplace policies), and the extent to which institutional messaging legitimizes and prioritizes public health risk in an environment of competing risks will set the tone for message acceptance. Counter to prior research, which suggested that public health messaging should focus on increasing severity and threat [6], our results suggest this approach will fall short of the desired response.

Mask tweets were dynamic and reflected many meanings and questions in the communication of COVID-19 risk over time. While hyperlinks contextualized tweets, mentions at public officials served to direct questions or blame, rhetorical questions reflected uncertainty and hashtags took on emotional nature as impacts of the real COVID-19 risk became more dire with time. The six derived themes identified reflect the multiple factors shaping risk perception dynamically in the sense-making process. Attention to social and group identity constructions of risk, normalizing public health recommendations through workplace policy and shifting away from individual responsibility in a politicized climate may prove advantageous for effective public health messaging. Our results indicate that citizens ascribed many meanings to masking and the simultaneous amplification and attenuation of COVID-19 risk complicates public health messaging. The social, psychological, and institutional experience of risk (behavior of others, norms, role modeling and messaging of government) come into play as much if not more so than objective risk perceptions and need to be taken into account in public health

messaging when the goal involves increasing compliance with recommended risk mitigation behaviors.

## Supporting information

**S1 List. Coronavirus keywords.**
(DOCX)

**S2 List. Top 150 news agencies filtered.**
(DOCX)

**S1 Table. Risk perception codebook.**
(DOCX)

## Author Contributions

**Conceptualization:** Suellen Hopfer, Chen Li, Gloria Mark.

**Data curation:** Yuwen Lu, Ted Grover, Quishi Bai, Yicong Huang, Chen Li.

**Formal analysis:** Suellen Hopfer, Emilia J. Fields, Yuwen Lu, Ganesh Ramakrishnan.

**Funding acquisition:** Suellen Hopfer, Chen Li, Gloria Mark.

**Investigation:** Suellen Hopfer, Chen Li, Gloria Mark.

**Methodology:** Suellen Hopfer, Emilia J. Fields, Yuwen Lu, Ganesh Ramakrishnan, Ted Grover, Yicong Huang, Gloria Mark.

**Project administration:** Yicong Huang, Chen Li.

**Resources:** Quishi Bai, Yicong Huang, Chen Li.

**Software:** Yuwen Lu, Ted Grover, Quishi Bai, Yicong Huang, Chen Li.

**Supervision:** Suellen Hopfer, Emilia J. Fields, Chen Li, Gloria Mark.

**Visualization:** Yuwen Lu, Ted Grover, Quishi Bai.

**Writing – original draft:** Suellen Hopfer.

**Writing – review & editing:** Emilia J. Fields, Ted Grover, Chen Li, Gloria Mark.

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
