## [Decision Letter · Decision Letter 0]

1 Jul 2021

PONE-D-21-17166

The Social Amplification of Covid-19 Risk Perception Shaping Mask Wearing Behavior: A Longitudinal Twitter Analysis

PLOS ONE

Dear Dr. Hopfer,

Thank you for submitting your manuscript to PLOS ONE. After careful consideration, we feel that it has merit but does not fully meet PLOS ONE’s publication criteria as it currently stands. Therefore, we invite you to submit a revised version of the manuscript that addresses the points raised during the review process.

We look forward to receiving your revised manuscript.

Kind regards,

Prof. Anat Gesser-Edelsburg, Ph.D.

Academic Editor

PLOS ONE

Journal Requirements:

Reviewers' comments:

Reviewer's Responses to Questions

**Comments to the Author**

1. Is the manuscript technically sound, and do the data support the conclusions?

Reviewer #1: Partly

Reviewer #2: Yes

Reviewer #3: Yes

Reviewer #4: Yes

2. Has the statistical analysis been performed appropriately and rigorously? 

Reviewer #1: No

Reviewer #2: Yes

Reviewer #3: Yes

Reviewer #4: Yes

3. Have the authors made all data underlying the findings in their manuscript fully available?

Reviewer #1: No

Reviewer #2: No

Reviewer #3: Yes

Reviewer #4: Yes

4. Is the manuscript presented in an intelligible fashion and written in standard English?

Reviewer #1: Yes

Reviewer #2: Yes

Reviewer #3: Yes

Reviewer #4: Yes

5. Review Comments to the Author

Reviewer #1: The paper represents a useful application of the social amplification risk framework to COVID-19 risk perception related to mask-wearing on the basis of a sample of messages on twitter. The qualitative review of themes and how the messages changed over time were interesting and seem well conducted. The way that the data was used to answer the research questions was clear, with both examples and graphs. However, the analysis in the paper suffered greatly from a lack of clear connection between the excellent research questions and data and the conclusions, and many conclusions were simply not supported by the analysis performed. Despite these issues, and other issues highlighted below, with further clarification and some additional work, this analysis will be a useful contribution to the literature on both COVID-19 risk perception, and social media analysis of public response.

Specific concerns and suggestions:

- Statements in the discussion about implications seem unconnected to the actual analyses. This is true in man places, but as one example, while it is very clear from other literature and experience that “...an information-deficit model approach to public health messaging... will unlikely be sufficient,” what part of the analysis justifies the claim in the paper that “Findings also suggest” this point? Other claims that seem unrelated to the analysis, which should be removed or more clearly tied to the analysis, include “group mask-wearing norms and policies normalizing mask wearing play critical roles,” and “Investment in monitoring social media sites as amplification stations for public health risk is warranted.”

- All of the figures would benefit from significantly more explanatory text, so that they can be read and understood without closely reading the paper. Additionally, figure 4 would be greatly improved if the time-intervals were clarified. There seems to be a surprising periodicity, and I can only assume this is because of (uneven) time-period binning. Otherwise, this should be explained.

Methodological questions:

- The low intercoder reliability seems to imply that the concepts and interpretations were unclear even to the coders. Is there a literature on twitter qualitative analysis that can provide comparative numbers in other studies, or some reason (other than lack of construct clarity or overlap) to think that this isn't a defect in the analysis? Also, the specific method used seems difficult. It seems some specific source or justification is needed for applying Fleiss's Kappa in this multi-categorical assignment case. (A statistician should be consulted.)

The dataset was filtered to remove tweets that only contained hashtags or user mentions. Does this exclude when a reply message is intended to highlight a tweet? Does this exclude quote-tweets with only a hashtag or mention?

The keyword list was clearly designed to catch non-English tweets, but then the tweets were geolocated and only English tweets were included. This should be explained. Several top news propagators are also clearly not-US based or used by US twitter accounts. This should be explained as well. (Presumably, the data collection is being used for other projects, which is fine, but should be stated)

Given the global nature of the risk and spread, it seems strange that discussion of masks in places other than the US should be automatically coded as non-relevant. Can you explain this choice?

Minor technical issues:

Data availability upon request is insufficient, as the guidelines state. Despite twitter terms of service restricting sharing the full dataset publicly, the tweet ids can be made fully public to allow reconstruction of the dataset without requesting data. If this is not done, a clear justification is needed. (Also, the availability statement also states that the codebook would be available upon request, but it is in fact included as appendix S3.)

The ethics statement should have a review number, or a date and the format in which the exemption was provided.

Reviewer #2: The manuscript is worthwhile for its qualitative summary and categorized of raw voices exchanged on twitter about U.S. people's reactions to issues related to Covid-19. In addition, a transition of the people's risk perception can be understood along a timeline, and there are interesting contextual considerations specific to qualitative research methods. While some of the conclusions, such as the need to build a presence and trust of public health officials delivering the messages and the importance of public figures role models, seem a bit lacking novelty, devising messaging according to the social and group identity of the recipient, rather than increasing the severity and threat is a new recommendation.

I would like to confirm a few points.

- How much influence does twitter have on what range of people compared to other media?

- Does the difference in expertise of the three coders affect the results of the analysis? It would be better to clarify the background of the coders.

- Although the overall appearance rate of “Covid desensitization” is low, why was it extracted as one category and the kappa coefficient remained low at 0.28?

- How does "tailoring public health messaging by psycho-behavioral profiles to effectively reach heterogenous subgroups regarding Covid-19 risk." in the discussion specifically refer to?

Reviewer #3: I wish to congratulate the authors to this thorough analysis of a new aspect of risk communication that derives very useful recommendation for public health risk communication. The application of the conceptual framework of SARF is a useful theory to undertake this very interesting investigation.

Reviewer #4: First of all, I would like to pay tribute to the authors for completing this study. The author used Twitter data to examine the social dynamics of risk perceptions regarding mask-wearing in the COVID-19 era. The author found seven categories of Tweet content, which seem to represent the history of American society's reaction to COVID-19. This study allows us to think about how risk perceptions are diffused in a crisis, which could be important findings for determining public health strategies. However, although I mostly agree with the publication of your manuscript, a few concerns remain.

1.

You defined risk perception as a communication process situating risk perception cognitively and socially, culturally, and historically. This definition was based on Kasperson’s discussion. Therefore, we could rephrase this definition to a communication process in which risk signals against COVID-19 interact with psychological, social, institutional, and cultural processes in ways that intensify or attenuate perceptions of risk and its manageability and shape risk behavior.

Based on the above definition, you discussed how the communication process of risk occurs in society by analyzing the content of tweets. However, was the analysis design of this study in ways that would allow you to explore such communication processes? Your analysis did not address the features of Twitter's communicable tools such as mentions, retweets, and replies and thus did not adequately discuss the relationships within and between the seven extracted themes. Is it possible that the results represent the size of the reaction to social events rather than describing risk perception as a communicative process? Please add a discussion of the communication process within or across themes with the limitations of this study's analysis.

2.

The process of risk perception includes intensify and attenuate perceptions. In fact, Figure 4 shows that the tweets for each risk perception theme repeatedly increase and decrease. However, your discussion was biased towards an increase in risk perception. It would be better if you also discussed attenuating risk perception.

These comments may contain my misunderstanding. If so, please point it out.

6. PLOS authors have the option to publish the peer review history of their article (what does this mean?). If published, this will include your full peer review and any attached files.

Reviewer #1: **Yes: **David Manheim

Reviewer #2: No

Reviewer #3: No

Reviewer #4: **Yes: **Tomoyuki Kobayashi

---

## [Author Response · Author response to Decision Letter 0]

18 Aug 2021

We thank the reviewers for their time and feedback. The now separate document as requested, the response to reviewers, includes a point by point response to all reviewer questions.

---

## [Decision Letter · Decision Letter 1]

1 Sep 2021

The Social Amplification and Attenuation of COVID-19 Risk Perception Shaping Mask Wearing Behavior: A Longitudinal Twitter Analysis

PONE-D-21-17166R1

Dear Dr. Hopfer,

We’re pleased to inform you that your manuscript has been judged scientifically suitable for publication and will be formally accepted for publication once it meets all outstanding technical requirements.

Kind regards,

Prof. Anat Gesser-Edelsburg, Ph.D.

Academic Editor

PLOS ONE

Additional Editor Comments (optional): Please address reviewer #1 minor comments during the proofreading stage.

Reviewers' comments:

Reviewer's Responses to Questions

**Comments to the Author**

1. If the authors have adequately addressed your comments raised in a previous round of review and you feel that this manuscript is now acceptable for publication, you may indicate that here to bypass the “Comments to the Author” section, enter your conflict of interest statement in the “Confidential to Editor” section, and submit your "Accept" recommendation.

Reviewer #1: All comments have been addressed

Reviewer #2: All comments have been addressed

2. Is the manuscript technically sound, and do the data support the conclusions?

Reviewer #1: Yes

Reviewer #2: Yes

3. Has the statistical analysis been performed appropriately and rigorously? 

Reviewer #1: Yes

Reviewer #2: Yes

4. Have the authors made all data underlying the findings in their manuscript fully available?

Reviewer #1: No

Reviewer #2: Yes

5. Is the manuscript presented in an intelligible fashion and written in standard English?

Reviewer #1: Yes

Reviewer #2: Yes

6. Review Comments to the Author

Reviewer #1: Thank you for your revisions and addressing the various concerns, especially the explanation about the statistical issues with various intercoder reliability measures. I believe the paper is an excellent contribution, and I'm happy to see the improvements which were made.

3 very minor points:

- The github repo, https://github.com/ISG-ICS/Geolocated_USA_tweets_w_mask_IDs, has a readme.md with several spelling errors in the title of the paper.

-The statement about the data is missing the word "not" - "According to Twitter agreement, we release IDs (Tweet and User ID) of the 7k mask tweets, but [NOT] their content."

- Line 816, the added word "and" seems like it should be removed, and on 817, I believe you want to add the word "are" - ", and [ARE] more likely".

Reviewer #2: #1 Thank you for your response. The representativeness of twitter users was confirmed in the limitation section.

#2 Thank you. It’s confirmed.

#3 As you answered, I think qualitative research is also about finding important events in a small number of cases, so I look forward to further research.

#4 The recommendations for public health messaging suggested by this study have been revised to make them easier to understand.

7. PLOS authors have the option to publish the peer review history of their article (what does this mean?). If published, this will include your full peer review and any attached files.

Reviewer #1: **Yes: **David Manheim

Reviewer #2: No

---

## [Editor Report · Acceptance letter]

15 Sep 2021

PONE-D-21-17166R1 

The Social Amplification and Attenuation of COVID-19 Risk Perception Shaping Mask Wearing Behavior: A Longitudinal Twitter Analysis 

Dear Dr. Hopfer:

I'm pleased to inform you that your manuscript has been deemed suitable for publication in PLOS ONE. Congratulations! Your manuscript is now with our production department. 

Kind regards, 

on behalf of

Prof. Anat Gesser-Edelsburg 

Academic Editor

PLOS ONE